# Forensic Application of Genetic and Toxicological Analyses for the Identification and Characterization of the Opium Poppy (*Papaver somniferum* L.)

**DOI:** 10.3390/biology11050672

**Published:** 2022-04-27

**Authors:** Roberta Tittarelli, Angelo Gismondi, Gabriele Di Marco, Federico Mineo, Francesca Vernich, Carmelo Russo, Luigi Tonino Marsella, Antonella Canini

**Affiliations:** 1Laboratory of Forensic Toxicology, Section of Legal and Forensic Medicine, Social Security and Forensic Toxicology, Department of Biomedicine and Prevention, Faculty of Medicine and Surgery, University Tor Vergata, Via Montpellier 1, 00133 Rome, Italy; federicomineo@yahoo.it (F.M.); francescavernich@hotmail.it (F.V.); carmelo.russo79@gmail.com (C.R.); marsella.luigi@gmail.com (L.T.M.); 2Laboratory of Botany, Department of Biology, Faculty of Mathematical, Physical and Natural Science, University Tor Vergata, Via della Ricerca Scientifica 1, 00133 Rome, Italy; gismondi@scienze.uniroma2.it (A.G.); gabriele.di.marco@uniroma2.it (G.D.M.); canini@uniroma2.it (A.C.)

**Keywords:** forensic toxicology, opium poppy, specie identification, *Papaver somniferum* L., GC-MS, genetic analysis, DNA barcoding, RAPD

## Abstract

**Simple Summary:**

The identification and characterization of illicit plant species such as *Papaver somniferum* L. (opium poppy) are essential in order for law enforcement agencies to obtain information about the origin of seized drugs, and to improve the monitoring of illegal drug trafficking routes. This study focuses on the genetic and toxicological analyses of some poppy capsules seized by the Italian Police Forces in 2017 and 2018. The detection of specific morphological traits, the chemical study of the alkaloid profile by gas chromatography coupled with mass spectrometry (GC/MS), and the genetic analysis performed by the DNA barcode approach showed that the seized poppy capsules belonged to *P. somniferum*. The data collected also suggested that the opium capsules originated from Afghanistan.

**Abstract:**

Background: A reliable and science-based taxonomic determination of *Papaver somniferum* L. (opium poppy), the illegal species of the genus *Papaver*, may have practical and legal implications for law enforcement. *P. somniferum* is a controlled plant because of its narcotic substances, such as morphine and codeine. As poppy plants have rather similar morphological features, both chemical and genetic analysis are required in order to achieve an accurate characterization of such species. The chemical structures of alkaloids are extremely variable even within the same species, which is why the genetic approach may lead to a more scientific *Papaver* sp. differentiation. The aim of our study was the taxonomic identification of poppy capsules seized by the Italian Police Forces being considered as potential *P. somniferum* derivatives. Methods: The alkaloids detected using gas chromatography/mass-spectrometry (GC/MS) were morphine, codeine, thebaine, noscapine, meconin, hydrocotarnine, and traces of papaverine. Further genetic analysis was carried out simultaneously using three plastid DNA barcoding regions (matK, trnH-psbA, and rbcL) for the samples’ identification. Results: The Random Amplification of Polymorphic DNA (RAPD) method showed that the analysed samples were genetically identical. Conclusions: The morphological, toxicological, and genetic profile of the samples revealed that they belonged to *P. somniferum* species. Furthermore, the alkaloid content of dried poppy capsules might be used to investigate and track their origin.

## 1. Introduction

The opium poppy (*Papaver somniferum* L.) is widely used for medicinal purposes due to the analgesic effects of its alkaloids (the narcotic analgesic, morphine; the cough suppressant, codeine; the pain reliever, thebaine; the muscle relaxant, papaverine; and the antitumoral agent, noscapine) [1,2,3].

The genus *Papaver* includes about 110 species [4]. Among *Papaver* subspecies, only *P. somniferum* and *P. setigerum* contain narcotic substances, including morphine, codeine and thebaine [5,6]. *P. setigerum* is characterized by significantly higher concentrations of papaverine and lower concentrations of morphine than *P. somniferum* [2,7,8,9].

The major alkaloid present in the opium poppy is generally morphine, followed by noscapine, codeine, thebaine and papaverine [2,10,11]. Some exceptions have been published in literature reporting noscapine, papaverine, or codeine as the major component of these subspecies [2,12]. This variation in morphine concentration may occur due to losses related to leaching, the fungal and enzymatic degradation of alkaloids, or wet and cold weather during maturation [2,13,14]. The variability of the alkaloid content in the opium poppy depends on several factors, such as its region of origin [15,16], climate conditions (e.g., temperature, light conditions) [16], plant cultivar [3,17], vegetation stage [3,18], and the different plant parts examined (the stem diameter, peduncle length, and plant height) [3]. Changes in the content of alkaloids are also derived from capsules’ sizes, the status of conservation (dry or fresh), the presence of seeds or incisions [1,15], and the number of incisions made to extract the opium.

Currently, the cultivation of opium poppy is closely monitored by law enforcement because it is an indirect source of heroin, one of the most dangerous narcotic substances, carrying a high risk of addiction and abuse [19].

According to the World Drug Report 2021, illicit opium poppy cultivation showed a peak in 2017, then a rebound in 2020, rising by 24% compared with the previous year and reaching 295,000 hectares [20]. These data also revealed that the main opium production areas were located in Asia, where 76% of all global opioid seizures were carried out in 2019 [20].

The United Nations Office on Drugs and Crime (UNODC) demands that Member States and the concerned agencies and organizations report and share all of their information about the origin of the seized narcotic substances.

Law enforcement agencies may obtain information about the origin of seized drugs from attendant circumstances such as the pathway of the finds; the presence of newspaper, wrappings, and labels; or the statements made by the traffickers themselves, although these data cannot always be considered as solid and reliable. Therefore, in order to obtain consistent information, the drug origin should be inferred from the intrinsic characteristics of the narcotics [21], so as to enable a better and more effective monitoring, and to fight against the illegal drug trafficking performed by the law enforcement agencies.

In order to determine the origin of opium poppy and its derivatives (i.e., opium and heroin), its morphine content is generally used. The morphine percentage in opium ranges considerably (from 3% to 30%) across countries, unlike its content in dry poppy capsules, which remains practically unchanged (from 0.2% to 1.0%) [16]. In addition to morphine, other alkaloids and their ratios compared to the morphine content can provide information about the origin. For example, the codeine concentration in opium from India, Iran, or Afghanistan is almost twice that from Turkey or the former Yugoslavia [15]. The presence of papaverine and narcotine (also known as noscapine) is typical of Afghan and Thai opium poppy [16]; in particular, a low concentration of papaverine is the distinctive feature of samples coming from Afghanistan and India [21]. Indian opium is also characterized by low thebaine concentrations [22] and relatively high cryptopine and porphyroxine (meconidine) concentrations, the dosages of which have been proposed as an analytical test to track Indian opium [21]. Furthermore, the geographical origin of heroin samples could be also investigated through the analysis of its manufacturing byproducts [23].

Some *Papaver* spp. are very difficult to identify solely on the basis of their morphological traits. In these cases, for a proper taxonomic classification [9,24,25,26], a worthy chemical and genetic approach is required.

Chemical analysis for species identification consists of the determination of the main alkaloids which are detectable in the plant material and the study of their ratios compared to known plant samples and standard alkaloids. Furthermore, in the case of illicit species detection, as in case of opium poppy, the presence of substances of abuse would confirm its illegality.

In general, for species identification, DNA analysis may surely provide unequivocal information. In particular, DNA barcoding represents one of the most innovative and modern techniques for species discrimination [27,28,29,30]. Indeed, studying the nucleotide succession of highly conserved genetic traits and comparing them with scientific databases, the botanical origin of DNA molecules extracted from both modern and ancient plant residues can be detected [31,32].

According to this premise, in our study, we taxonomically classified poppy capsules by their phenotypic features. Moreover, we confirmed the morphological prediction using a chromatographic approach for the determination of the alkaloid profile of the samples, and using a DNA barcoding technique using three plastid DNA regions: the maturase K gene (matK), the intragenic spacer between the tRNAHisGUG gene and the photosystem II thylakoid membrane protein of the Mr 32,000 gene (trnH-psbA), and the ribulose 1,5-bisphosphate carboxylase/oxygenase large subunit gene (RuBisCO large subunit, rbcL). Finally, the existence of genetic differences among the different samples was assessed by the RAPD (Random Amplification of Polymorphic DNA) method.

## 2. Materials and Methods

### 2.1. Morphological Analyses of Plant Material

The National Police in the south metropolitan area of Rome seized two sets of trafficking samples of poppy capsules in 2017 and 2018. The 2017 seizure consisted of 254 envelopes containing dried opium bulbs with a total gross weight of 26.4 kg. Each envelope contained capsules with a gross weight of approximately 105 g, in most cases fragmented into small pieces. The 2018 seizure consisted of 279 envelopes with a total gross weight of 29.6 kg, with packaging characteristics very similar to the previous one: each envelope contained capsules with a gross weight of approximately 106 g, and the material was more fragmented than that in the previous seizure. For the purposes of our study, only the whole capsules were subjected to analysis.

From each seizure, eight whole capsules were collected and analyzed. The capsules were emptied and deprived of their latex product.

The plant material was placed on millimetric paper to be photographed [Figure 1]. The dimensions, shapes and structural characteristics of the samples were compared to those reported in the literature [33], in order to identify their botanical origin at the species level.

### 2.2. Chemical Analysis

#### 2.2.1. Instrumentation

The quantitative analyses were performed by gas chromatography with the flame ionization detection (GC-FID) method in order to assess the percentage of morphine purity, using ethaverine as an internal standard (ISTD).

The GC-FID instrumentation consisted of an Agilent 7820A GC system and an Agilent G4513A series auto sampler (Agilent Technologies, Palo Alto, CA, USA). The column was an Agilent HP-5 fused silica capillary column 30 m in length, 0.32 mm ID, and with a 0.25 μm film thickness (Agilent Technologies, Palo Alto, CA, USA). The carrier gas (N_2_) flow was constant at 1 mL/min. A total of 1 μL of each sample was injected into the gas chromatography system using a 5:1 split injection ratio. The injector temperature was 290 °C. The oven temperature was set as follows: the initial temperature of 200 °C was held for 0.5 min, then increased to 260 °C with a rate of 15 °C/min, and held for 4 min. The total run time was 8 min.

The qualitative analyses, carried out to identify the alkaloid content, were performed by gas chromatography coupled with mass spectrometry (GC/MS).

The GC/MS instrumentation consisted of a 7890A GC system equipped with a 7683B autosampler and interfaced to a single quadrupole 5975C mass-selective detector (MSD, Agilent Technologies, Palo Alto, CA, USA). The GC column was an Agilent HP-5MS fused silica capillary column (30 m long, 250 μm ID, and 0.25 μm film thickness). The carrier gas (He) flow was constant at 1 mL/min. A total of 1 μL of each sample was injected in pulsed splitless mode into the GC/MS, and the acquisition mode was set at a full scan with a mass scan range of 40–550 *m*/*z*. The MSD operated in electron ionization mode at 70 eV, and the detector voltage was 0.9 kV.

The injector port and the detector temperatures were set at 270 °C, and the transfer line was set at 280 °C. The oven temperature program was set to an initial temperature of 120 °C for 1 min, followed by an increase to 300 °C at 15 °C/min, which was held for 5 min. The total run time was 18 min.

The identification of the alkaloid contents was performed by comparison with the mass spectra reported in the NIST14 Mass Spectral Library.

#### 2.2.2. Standard Solutions and Reagents

Stock solutions of methanol, ethaverine and bis-trimethylsilyltrifluoroacetamide (BSTFA) were obtained from Merck (Darmstadt, Germany). Certified reference materials (CRM) of morphine and codeine solutions were purchased from Lipomed AG (Fabrikmattenweg 4, CH-4144 Arlesheim, Switzerland).

#### 2.2.3. Sample Preparation for Chemical Component Analysis

The opium capsules were opened and emptied of their contents, which were crushed to make samples ranging from 100 to 130 mg.

The quantitative analyses by GC-FID were performed by dissolving the powder into a methanol solution containing ethaverine 1 mg/mL. The dissolved samples were incubated for 20 min at room temperature, and then 1 μL of each sample was injected into the GC-FID.

After the poppy pulverization, 1 g of each powdered sample was collected into a Pyrex glass tube. In order to detect morphine in its free state, acid hydrolysis was performed with 2 mL of HCl 0.1 N [8], by heating the samples in a thermoblock at a controlled temperature (45 °C) overnight. After cooling, the samples were centrifuged at 4500 rpm for 5 min, and the supernatants were collected and adjusted to pH 10 by the addition of 1 M NaOH solution. The samples were then extracted by the addition of 0.5 mL ethyl acetate for 20 min using a rotating plate stirrer. After the centrifugation (at 4500 rpm for 5 min), the organic layers were evaporated to dryness under a gentle stream of nitrogen. The residues were reconstituted with 100 µL ethyl acetate, and 1 µL was injected for the qualitative GC/MS analysis of the alkaloids.

### 2.3. Genetic Analysis

In order to confirm the morphological and toxicological prediction, DNA barcoding analysis was carried out in compliance with our laboratory protocols [31,34,35].

#### 2.3.1. DNA Extraction

DNA was purified from the seeds present in the capsules (both from 2017 and 2018 samples), after pulverization using a mortar, pestle, and liquid nitrogen. In detail, DNA was extracted from the plant powder (20 mg) by a NucleoSpin Plant II Mini Kit (Macherey-Nagel Düren, Germany), following the manufacture’s guidelines. The nucleic acid was visualized under UV light (GelDoc 2000, BIO-RAD, Hercules, CA, USA), after electrophoretic separation on 1% agarose gel containing 10 mg/mL ethidium bromide (the molecular weights were MW1-MassRuler-MW1 and MW2-GeneRuler by Thermo Fisher Scientific (Waltham, MA USA).

#### 2.3.2. DNA Barcoding

The PCR reactions were performed using a BIO-RAD IQ5 iCycler, in a final volume of 50 μL, which consisted of 25 ng template; 2.5 U KAPA Taq DNA Polymerase (Kapa Biosystems); 20 μM of both primers (Sigma-Aldrich) for matK, rbcL and trnH-psbA genes (primer sequences reported in [31,35]); 0.2 mM for each dNTP; and 1X Taq DNA Polymerase Buffer. During the amplifications, the thermocycler was set as follows: (a) initial denaturation at 94 °C for 5 min; (b) 45 cycles of denaturation at 94 °C for 20 s, annealing at the appropriate temperature (56 °C for rbcL and matK; 54 °C for trnH-psbA) for 25 s and extension at 72 °C for 45 s; (c) final extension at 72 °C for 10 min; (d) storage at 4 °C. For the sequencing, the PCR products (2 μL) were first treated for 15 min at 37 °C with 0.5 μL ExoSAP-IT and 4.5 μL sterile distillated water, and were then supplemented with 1.5 μL Bright Dye Terminator (iCloning) and 1.5 μL of the corresponding primer (forward or reverse). This mix was subjected to 25 cycles of PCR, each one as reported: 95 °C for 10 s, 50 °C for 5 s, and 64 °C for 4 min. The labeled sequences were precipitated by adding 2 μL of 3 M sodium acetate (pH 5.2) and 50 μL of 95% cold ethanol. The samples were centrifuged for 30 min at 2000 rpm, and the pellets—washed twice with 150 μL of 70% ethanol—were resuspended in 20 μL of 100% formamide. The DNA was sequenced using a 3130 Avant Genetic Analyzer (HITACHI, Applied Biosystems), and the nucleotide successions were visualized by BioEdit v7.0.5 software.

#### 2.3.3. RAPD Analysis

For the RAPD analysis, in a final volume of 50 μL, 50 ng of total DNA were joined with 2.5 U KAPA Taq DNA Polymerase (Kapa Biosystems, Merck Life Science s.r.l., Milan, Italy), 20 μM of each random primer (Merck Life Science s.r.l, Milan, Italy), 0.2 mM for each dNTP, and 1X Taq DNA Polymerase Buffer. The random primers predicted by Operon Technologies and used in this study were: in the RAPD A assay, R1: 5′-CAGGCCCTTC-3′ + R2: 5′-TGCCGAGCTG-3′ + R3: 5′-AGTCAGCCAC-3′ + R4: 5′-AATCGGGCTG-3′; in the RAPD B assay, R5: 5′-AGGGGTCTTG-3′ + R6: 5′-GGTCCCTGAC-3′ + R7: 5′-GAAACGGGTG-3′ + R8: 5′-GTGACGTAGG-3′; and in the RAPD C assay, R2 + R4 + R6 + R8. The PCR amplifications were carried out in a BIO-RAD IQ5 iCycler according to the following parameters: (i) initial denaturation at 95 °C for 4 min; (ii) 60 cycles of denaturation at 95 °C for 20 s, annealing at 48 °C for 25 s, and extension at 72 °C for 45 s; (iii) final extension at 72 °C for 10 min; (iv) storage at 4 °C. The RAPD profiles were detected by UV light after fractionation on 1.5% agarose gel. In order to avoid any external contamination, all of the procedures were carried out as suggested in Gismondi et al. [32].

#### 2.3.4. Statistics

For both types of samples (2017 and 2018), three different sub-samples were considered, and each one was submitted for all of the previous tests in triplicate.

## 3. Results

### 3.1. Morphological Features

The plant material was photographed and analyzed. Both the 2017 and 2018 samples were identified as capsules of the *Papaver* genus, due to their peculiar shape (Figure 1A,B). The dehiscent dry fruits were different in size; on average, they were 3.5 cm in width and 5.4 cm in height. However, all of the samples could be morphologically attributed to *P. somniferum* L., according to Pignatti’s atlas [33]. Indeed, each of them showed phenotypic traits typical of this species: (i) a hairless sub-spherical-fluted capsule; (ii) fruit dimensions ranging from 3 to 5 cm in width and 4 to 8 cm in height; (iii) a stigmatic disk with more than 10 radiating rays; and (iv) a double ring joint on the peduncle. The seeds had a black–purplish color; this characteristic suggested that the samples belonged to the *glabrum* variety, which is commonly cultivated in Asia Minor [36].

### 3.2. Chemical Component Analysis

Quantitative analysis revealed that the morphine content of the samples ranged from 0.3 to 0.8% (average 0.64%) [Figure 2].

The morphine percentage is strictly related to the opium moisture content, which is usually higher (about six percent) in the case of drying at room temperature.

The quantity of morphine produced by the poppy is strictly related to the frequency of the incisions on the capsules. In several countries (such as Iran, India, and Afghanistan), the fruits are cut up four or five times per day, until they no longer produce latex. Obviously, the decrease of the yield of latex corresponds to a lower morphine content [20,37,38].

The poppy capsules, as shown in Figure 1, had several incisions, likely to collect the latex oozed from the plants, such that the morphine concentration found in the analyzed samples could be compatible with the practice described above.

The alkaloids isolated in the alkaline extracts obtained after acid hydrolysis are listed in Table 1.

A representative image of the total ion chromatogram concerning the poppy alkaloids extracted from the samples is shown in Figure 3. Codeine and thebaine were the most abundant alkaloids. Meconin, hydrocotarnine, neopine, laudanosine, and noscapine were also detected, together with traces of papaverine.

The identification of the compounds was performed by comparing them with reference compounds in the NIST 14 mass spectral library database.

### 3.3. Interpretation of the Genetic Data

In order to validate the morphological identification, the DNA barcoding technique was applied. The total DNA was extracted from seeds collected from within the capsules (Figure 3) and PCR amplifications of three barcode genes (matK, rbcL and trnH-psbA) were carried out. The amplicons were sequenced and compared to the GenBank nucleotide database. In Table 2, the nucleotide succession obtained for each barcode region of the DNA sample (query) and the GenBank accessions shows the highest values of sequence matching with the respective queries reported.

We observed that both the 2017 and 2018 sample groups showed the same barcode sequences, except for the presence of a single nucleotide polymorphism detected in the matK gene in position 413 (where a Guanine was substituted by an Adenine in the 2018 sample). No difference was found in the sequences of the three independent replicates for each sample. The intersection of the lists of GenBank accessions obtained through DNA barcoding analysis, as described in Gismondi et al. [31], corroborated our previous morphological prediction; our samples consisted of the fruits and seeds of *P. somniferum*.

In order to further check the similarity of the 2017 and 2018 plant DNAs (Figure 4), three RAPD assays (A, B and C) were performed. As is detectable in Figure 5, the RAPD profiles (6 bands in assay A, 8 in assay B, and 9 in assay C) of the two samples always showed the same patterns, indicating that their genomes were identical.

## 4. Discussion

The detection of specific morphological traits from the plant samples led us to identify them as capsules and seeds belonging to *P. somniferum* (Figure 1a,b). By GC/MS analysis, morphine, codeine, thebaine, noscapine, meconin, hydrocotarnine and traces of papaverine were detected. This alkaloidal profile corresponded to that expected for *P. somniferum*, as described in the literature [4,11,39].

The morphological and chemical results were also confirmed by the DNA barcoding approach, by sequencing three different plastid gene markers and comparing them with the GenBank nucleotide database (Figure 2 and Table 1). In our work, the discrimination ability of each barcode gene compared with the sequences available on GenBank appeared to be differential, if we considered the amplicons’ length and the specificity of the sequenced nucleotide successions. However, these aspects did not influence the data and the identification process reliability.

Accordingly, the two samples (2017 and 2018) appeared phenotypically, chemically, and genetically identical. Indeed, they presented the same molecular profiles even when submitted for RAPD assays, although three different primer moistures were applied (Figure 3). At the genetic level, just one deviation resulted from the two samples; more accurately, only a single nucleotide polymorphism showed deviation at the 413 position in the matK gene. However, this mutation can be considered to be a spontaneous phenomenon (naturally occurring at intraspecific level), as transition events occur more frequently than transversion ones due to the simpler molecular mechanisms involved in their generation resulting in silent aminoacidic substitutions without any crucial alterations [40,41].

The alkaloid profiling of poppy capsules could be used as a geographical origin indicator, although the large number of factors affecting both the content and typology of these metabolites could preclude clear results. Nevertheless, the alkaloid content may be considered as a potential indicator for origin identification. The alkaloid composition of the samples—characterized by the presence of noscapine, low papaverine concentrations and the absence of other typical poppy alkaloids—suggested an Afghan origin for the seized poppy capsules. Meconin and hydrocotarnine were also detected in the samples, but any direct correlation with the country of origin has not yet been reported in the existing literature. This composition was similar to that described by Remberg et al. regarding the alkaloid content of Afghan opium [38].

According to their research line, the labels “2017” and “2018” of the samples may indicate the harvest year. As reported above, the fact that the two series of samples were genetically identical confirmed their probable common origin from the same region.

Furthermore, if we consider, on the one hand, the significant impact that temperature and weather conditions have on the alkaloid profile of *P. Somniferum* during the harvest time (July) and, on the other, the chemical similarity between the two sample groups, we can suppose a cultivation zone with similar climate and environmental conditions both in July 2017 and July 2018 [16].

Thus, according to our data, the geographic origin determination of dry poppy capsules may have some limitations, not only because the incised poppy capsule contains lower alkaloid content concentrations than opium, but also because no distinctive origin-related trait can be detected. As heroin manufacturing may differ across countries in terms of reagents and methods, the manufacturing by-product analysis may confirm the country of production.

Even if the results obtained in our study were extremely promising, further studies on microsatellite STR analysis (STRs) are required in order to empower the proposed model of discrimination, although these genetic markers might be more reliable for distinguishing individuals belonging to the same species, rather than to different ones.

Moreover, in the near future the aim will be to validate the GC/MS analytical method in order to improve the quality of the study through the standardization of the produced data.

## 5. Conclusions

Chemical analysis for the geographical indication of origin of seized opium poppy is essential for law enforcement; however, this determination is still at a rather rough stage. Further research is needed in this field, including the creation of a dataset with authentic (known origin) samples, which can be investigated and classified on the basis both of their chemical and genetic profile.

## Figures and Tables

**Figure 1 biology-11-00672-f001:**
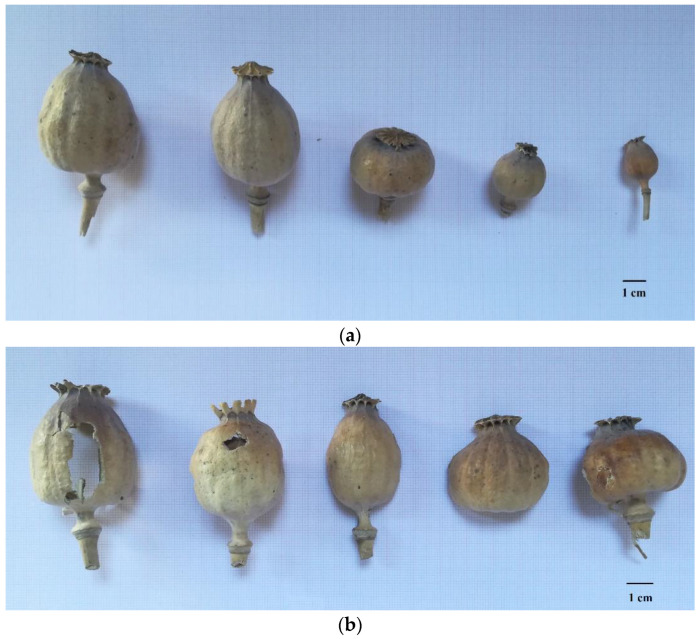
Images of the plant material on millimetric paper (the black bars correspond to 1 cm): representative samples of the sets seized in 2017 (**a**) and in 2018 (**b**).

**Figure 2 biology-11-00672-f002:**
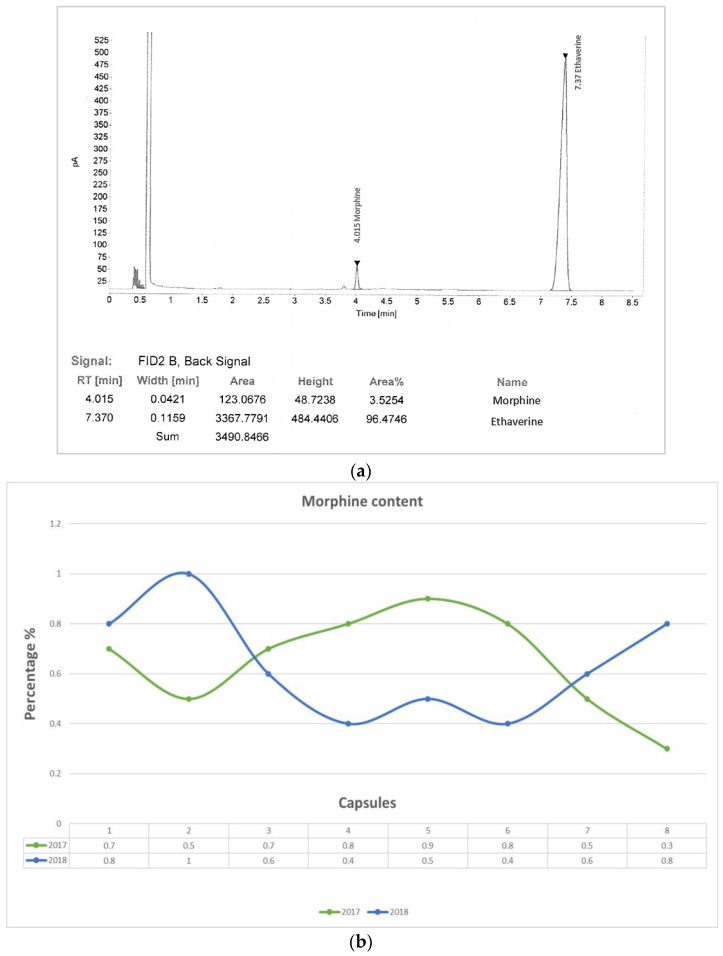
GC-FID chromatogram of the poppy extract (**a**) and a graphical representation of the morphine content variability in the analyzed samples (**b**).

**Figure 3 biology-11-00672-f003:**
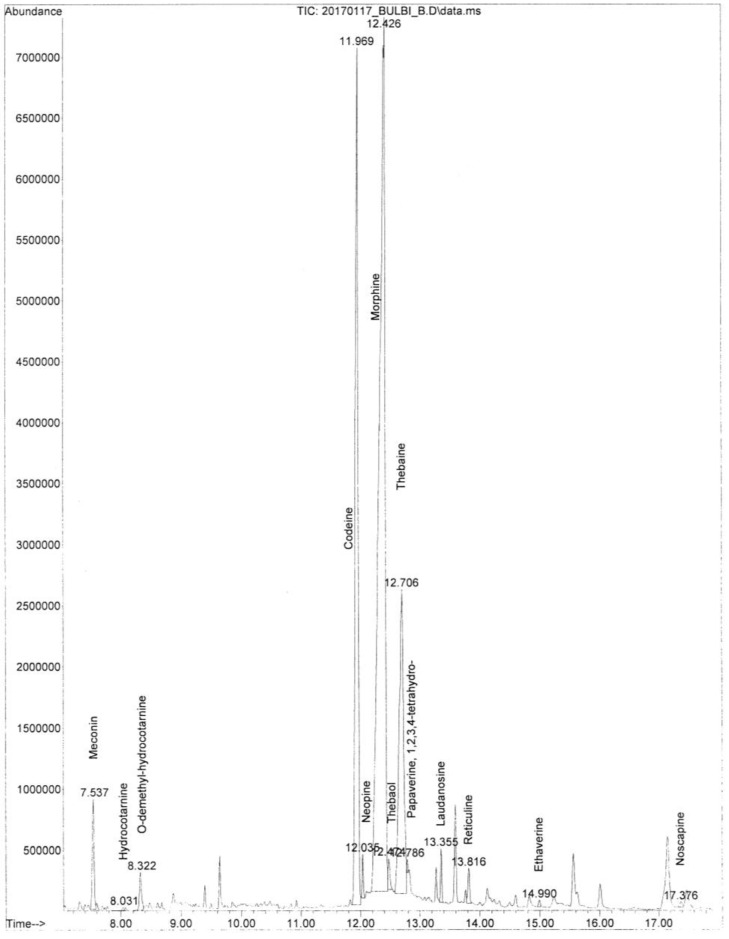
GC/MS total ion chromatogram of the alkaloids detected in the samples, including the ethaverine used as an internal standard.

**Figure 4 biology-11-00672-f004:**
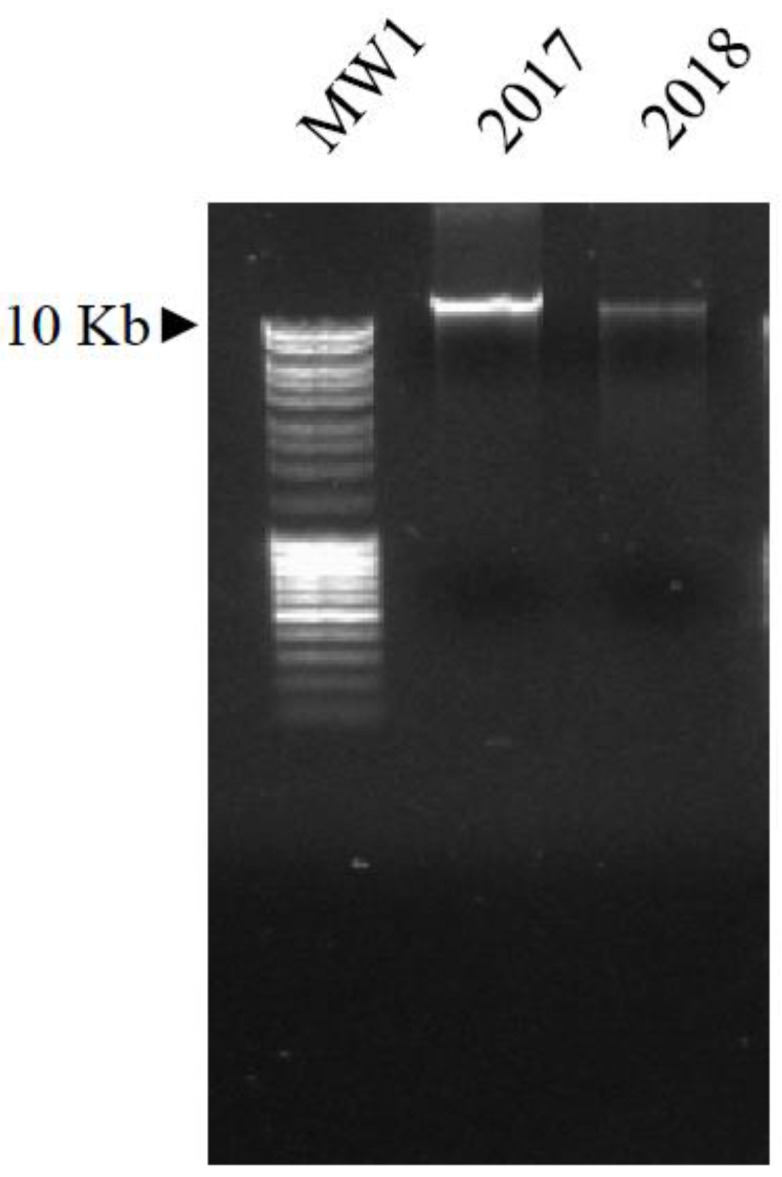
Visualization of the total DNA extracted from the two sets of samples (2017 and 2018), after separation on 1% agarose gel and exposure to UV light. MW1: molecular weight 1, as reported in the Materials and Methods section.

**Figure 5 biology-11-00672-f005:**
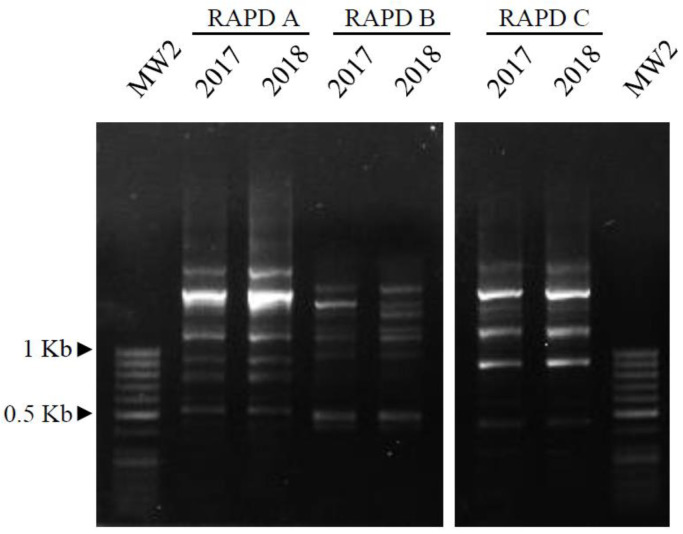
RAPD profiles of the two samples, obtained using three different primer mixes (assays A, B and C), after separation on 1.5% agarose gel and exposure to UV-light. MW2: molecular weight 2, as reported in the Materials and Methods section.

**Table 1 biology-11-00672-t001:** Alkaloids found in the opium poppy capsules.

Alkaloid	Retention Time (minutes)	Ions
Meconin	7.54	165-194(80)-147(70)-176(40)
Hydrocotarnine	8.00	220-178(70)-221(50)-205(30)
*O*-demethyl-hydrocotarnine	8.32	164-206(90)-207(50)-106(20)
Codeine	11.97	299-162(40)-229(30)-115(20)
Neopine	12.03	299-254(90)-284(20)-255(30)
Morphine	12.43	285-162(30)-215(30)-124(20)
Thebaol	12.47	254-239(90)-139(60)-127(40)
Thebaine	12.71	311-296(70)-268-(20)327(10)
*Papaver*ine 1,2,3,4-tetrahydro-	12.78	192-176(10)-206(10)-343(5)
Laudanosine	13.35	206-190(20)-207(20)
Reticuline	13.82	192-177(25)-193(10)
Ethaverine (ISTD)	14.99	366-395(60)-367(30)-394(25)
Noscapine	17.38	220-205(20)-221(20)

**Table 2 biology-11-00672-t002:** DNA barcoding analysis.

**matK SEQUENCE (614 bp)**	
AGCCGGCTTACTAATGGGATGCCCTGAAACGTTACAAAATTTCGTTTTAGCCAACGACCCAACCAAAGAAATAATTGGGACTTTGGTATCAAACTTCTTAATACAAATATCCATTATAAATGTACTTTCTATCATTTGACTCCTTACCACCGAAGGGTTAAGTCGTACACTTGCAAGATAGCTCAAAAGCTCGAGGGAATGATTGGATAATTTATTTATCTTAATTCTATCTGGTTGAGACCACAAGGAAAAATTACATTGCCATAAATTTACAAGATTCCATTTCCATTTATTCATCAAAAGAGTACTTCCCTTTGAAGCCAGAATTGATTTTCCTTGATATCTGACATAATGCATAAAAGGATCCTTGAACAACCATAGGAAGGTCTGAAAATCATTACTAAACACTACTGCAGGATGCTCCATTTTTCCATAGAAATTTATTCGCTCAAGAAGGATTCTAAAAGATGTTGATCGTAAACGAAAAGATTGTTTACGAAGAAAAACTAATATGGATTCGCATTCATATACATGAGAATTATATAGGAAGAAGAAAAAGCGTTGATTCTCCTTTGAAAAAAAAGGGAAATTGCCTTTATATTTTGAAAGTATTA	
**GenBank best results** (max score: 1077, total score: 1077, query cover: 98%, E-value: 0.0, Identity: 99%)	
**Description**	**Accession**
*Papaver rhoeas*	NC_037831.1
*Papaver somniferum* voucher QRI 504	MH287273.1
*Papaver rhoeas*	MF943221.1
*Papaver somniferum* voucher JK 028	MG221004.1
*Papaver rhoeas* voucher Hosam00241	KX783729.1
*Papaver somniferum*	KU204905.1
*Papaver somniferum* voucher SBB-0468	JN114767.1
*Papaver rhoeas* subsp. *rhoeas*	HE966959.1
*Papaver rhoeas* isolate NMW601	JN896019.1
*Papaver somniferum* isolate NMW602	JN896016.1
*Papaver rhoeas* isolate NMW3961	JN895411.1
*Papaver somniferum* isolate NMW3962	JN895410.1
*Papaver somniferum* isolate NMW3963	JN895409.1
*Papaver rhoeas* isolate NMW3959	JN894132.1
*Papaver rhoeas* isolate NMW5261	JN893897.1
*Papaver somniferum* subsp. setigerum	HM851028.1
*Papaver rhoeas* voucher USDA PI533721	GQ248175.1
*Papaver rhoeas*	FJ626525.1
**rbcL SEQUENCE (546 bp)**	
CCTGACGAGATGCTCCTCAACCTGGAGTTCCACCTGAGGAAGCAGGGGCCGCGGTAGCTGCCGAATCTTCTACTGGTACATGGACAACTGTGTGGACCGATGGACTTACCAGCCTTGATCGTTACAAAGGAAGATGCTACGACATCGAGCCCGTTGCTGGAGAAGACAATCAATATATTTGTTATGTAGCTTATCCTTTAGACCTTTTTGAAGAAGGTTCTGTTACTAACATGTTTACTTCCATCGTGGGTAATGTATTTGGGTTCAAAGCGCTTCGTGCTCTACGTCTGGAGGATCTGCGAATTCCTGTTGCTTATGTTAAAACTTTCCAAGGACCACCTCACGGTATCCAAGTTGAAAGAGATAAATTGAATAAGTATGGTCGTCCCCTATTGGGATGTACTATTAAACCAAAATTGGGGGTTATCTGCTAAGAACTACGGTAGGGCGGTTTATGAATGTCTACGTGGTGCACTTTGATTTTTACCAAGGGATGATGAAAAACGTGAACTCACAACCCTTTTATGCGTTGGAGAGATCGATTTC	
**GenBank best results** (max score: 931, total score: 931, query cover: 99%, E-value: 0.0, Identity: 98%)	
**Description**	**Accession**
*Papaver somniferum* voucher QRI 509	MH287278.1
*Papaver somniferum*	KU204905.1
*Papaver somniferum* subsp. *setigerum*	HM850231.1
**trnH-psbA SEQUENCE (312 bp)**	
TATATTTAATTTCTATATCACTCAAGGTTAGATATTTGAGTAGTTATCTATTAACTTTATTAATACTTAAATAAGTATAAGTATGTTGTACAAAAAAAGTAAATCCTTTCAATAAAAGGTACACTTTTTTATGGAAATAAAACAATACTAAAACTAAATGAAGGAGCAATACCGACCCTCTTATTCTATCAAGAGGGTCGGTATTGCTCCTTCAACTTCAACGCTTCATATACACTAAGACGGAAGTCTTATCCGTTTGTGGATGGAGCTTCAACAGCAGCTAGGTCTAGAGGGAAGTTGTGAGCATTACGT	
**GenBank best results** (max score: 564, total score: 564, query cover: 99%, E-value: 7e-157, Identity: 99%)	
**Description**	**Accession**
*Papaver somniferum*	KU204905.1
*Papaver somniferum*	JN584668.1

## Data Availability

The data presented in this study were obtained from the included studies, and are openly available.

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
