# Peer review of "Forensic Application of Genetic and Toxicological Analyses for the Identification and Characterization of the Opium Poppy (Papaver somniferum L.)"

_biology, 2022, doi:10.3390/biology11050672_

Round 1
Reviewer 1 Report
Review on the manuscript of Tittarelli R. et al.: “Forensic application of genetic and toxicological analyses for the identification and characterization of opium poppy (Papaver somniferum L.)”.
This manuscript explores different approaches to classify poppy capsules seized by the Police Forces in Italy into species of the Papaver genus. The authors used phenotypic characteristics and the profile of alkaloids (based on gas chromatography approaches) to taxonomically classified poppy capsules. In addition, the authors analyzed 3 genomic regions (maturase K gene, intragenic spacer between tRNAHisGUG and trnH‐psbA genes, and the ribulose 1,5‐bisphosphate carboxylase/oxygenase large subunit gene) and differences between samples were further assessed by a random amplification of polymorphic DNA method.
The data shown in the manuscript seem to be clear, and, in general, well described. However, some issues that arise in the manuscript, which need to be clarified in more detail, are listed below for consideration of the authors.
1 – Based on some issues I found all over the manuscript, I would recommend the authors to have their work proofread by an English native speaker. It would help the overall quality of the work.
2 – In figure 1, could the authors indicate what the scale bar means?
3 – In figure 2, the authors just show a chromatogram, where we can identify morphine and the internal standard (ethaverine). I would recommend the authors to include in this figure, together with the chromatogram, the graphical representation of morphine content in the different samples. It would help in understanding how variable the morphine content is between samples.
4 – In figures 4 and 5, I would recommend the authors to cut the gel images in a way that the wells are not visible.
5 – In order to have a solid conclusion on the origin of the samples (the authors suggest an Afghan origin), is it possible to obtain poppy capsules from Afghan and then compare the profile with the samples under study? It would help in establishing a clear‐cut conclusion.
Author Response
1 – Based on some issues I found all over the manuscript, I would recommend the authors to have their work proofread by an English native speaker. It would help the overall quality of the work.
Response 1: Thank you for the advice. The work has been proofread, as you suggested, by an English native speaker.
2 – In figure 1, could the authors indicate what the scale bar means?
Response 2: We added in the figure the value in centimeters corresponding to the length of the bar.
3 – In figure 2, the authors just show a chromatogram, where we can identify morphine and the internal standard (ethaverine). I would recommend the authors to include in this figure, together with the chromatogram, the graphical representation of morphine content in the different samples. It would help in understanding how variable the morphine content is between samples.
Response 3: A graph of the morphine content has been included in figure 2.
4 – In figures 4 and 5, I would recommend the authors to cut the gel images in a way that the wells are not visible.
Response 4: The gel images were cut as you rightly suggested.
5 – In order to have a solid conclusion on the origin of the samples (the authors suggest an Afghan origin), is it possible to obtain poppy capsules from Afghan and then compare the profile with the samples under study? It would help in establishing a clear‐cut conclusion.
Response 5: Currently, it's not possible to get opium capsules from Afghanistan because they are samples whose handling and shipping require special permits and very long delivery times.
Thank you very much for your comments.
Reviewer 2 Report
It is very interesting, well written and structured article. There is both scientific and forensic interest which is combined with application of Law. Some minor corrections should be done.
Line 89: I believe, nowadays, the word Yugoslav has to be changed or clarified, since typically this country is not included in Europe as a separate one.
Line 100: Printing mistake. It has to be written consists of instead in
Line 145: I think 30 m length is a printing mistake. Please change with correct mm (probably).
Line 157: Same as line 145
Line 208: Same as line 145
2.2.3 : You have to refer exactly how many opium capsules were opened and which was the average weight from the total number of packs of the selected species. It is critical to write the number of capsules since there are a lot of deviations of content during manufacturing, filling and packaging.
Line 221: 2000 g is rather a printing mistake. Probably, you mean mg.
Table 1: the title has to be above the table. So, transfer it properly.
Discussion: You have to write, definitely, that the method is not validated and has to be in the future.
Author Response
Line 89: I believe, nowadays, the word Yugoslav has to be changed or clarified, since typically this country is not included in Europe as a separate one.
Response 1: Thank you for your comment. We changed the word in the proper way (line 94→ the former Yugoslav).
Line 100: Printing mistake. It has to be written consists of instead in.
Response 2: Thank you. We changed the text as you suggested (line 105).
Lines 145/157 and 208: I think 30 m length is a printing mistake. Please change with correct mm (probably).
Response 3: there are no printing mistakes, as 30 meters is the column lenght of the GC system.
2.2.3 : You have to refer exactly how many opium capsules were opened and which was the average weight from the total number of packs of the selected species. It is critical to write the number of capsules since there are a lot of deviations of content during manufacturing, filling and packaging
Response 4: Thank you for your comment. In the paragraph we added a description for each seizure, adding some information about their weight and composition.
Line 221: 2000 g is rather a printing mistake. Probably, you mean mg.
Response 5: it was no a printing mistake. We were referring to wthe centrifuge revolutions per minute. We changed the unit of measure from g to rpm (line 236).
Table 1: the title has to be above the table. So, transfer it properly.
Response 6: Thank you. We transferred it as you suggested.
Discussion: You have to write, definitely, that the method is not validated and has to be in the future.
Response 7: in the final discussion according to your comment we added the need to validate the analytical method.
Thank you for your helpful comments.